# How to choose the size of facial areas of interest in interactive eye tracking

**Antonia Vehlen📷, William Standard, Gregor Domes***

Department of Psychology, Biological and Clinical Psychology, University of Trier, Trier, Germany

* domes@uni-trier.de

**Data Availability Statement:** The data sets created and analysed in this study can be downloaded here: https://osf.io/ytrh8/ or reproduced using the software tool described, which can also be downloaded from this website.

## Abstract

Advances in eye tracking technology have enabled the development of interactive experimental setups to study social attention. Since these setups differ substantially from the eye tracker manufacturer's test conditions, validation is essential with regard to the quality of gaze data and other factors potentially threatening the validity of this signal. In this study, we evaluated the impact of accuracy and areas of interest (AOIs) size on the classification of simulated gaze (fixation) data. We defined AOIs of different sizes using the Limited-Radius Voronoi-Tessellation (LRVT) method, and simulated gaze data for facial target points with varying accuracy. As hypothesized, we found that accuracy and AOI size had strong effects on gaze classification. In addition, these effects were not independent and differed in falsely classified gaze inside AOIs (Type I errors; false alarms) and falsely classified gaze outside the predefined AOIs (Type II errors; misses). Our results indicate that smaller AOIs generally minimize false classifications as long as accuracy is good enough. For studies with lower accuracy, Type II errors can still be compensated to some extent by using larger AOIs, but at the cost of more probable Type I errors. Proper estimation of accuracy is therefore essential for making informed decisions regarding the size of AOIs in eye tracking research.

## Introduction

Eye tracking, especially in its video-based form, has become a standard method for investigating visual attention in many research areas, including social neuroscience [1, 2], psychopharmacology [3–5] and virtual reality [6, 7]. In recent years, several studies have questioned common reporting standards in this field, or the lack of data quality evaluations regarding the specific setup in use [8, 9], e.g., some studies only report the overall accuracy of hardware determined by the manufacturer under ideal test conditions. The issue of data quality and inadequate reporting standards seems to be increasingly relevant as the field advances to develop more naturalistic, interactive or face-to-face eye tracking applications [10–12]. Not only do these setups deviate further in design from the manufacturer's test conditions, but when used in naturalistic interactions, other factors can affect the accuracy, such as movements accompanying facial expressions, speech or varying viewing distances.

Area of interest (AOI) based gaze classification is a popular approach for analyzing gaze data [13, 14]. This approach determines whether a gaze point or a fixation is directed at a

**Funding:** The study was in part supported by grants from the German Research Foundation (DO1312/5-1) to GD and the Trier University Research Priority Program "Psychobiology of Stress", funded by the State Rhineland-Palatinate. The funders had no role in study design, data collection and analysis, decision to publish, or preparation of the manuscript.

**Competing interests:** The authors have declared that no competing interests exist.

predefined region around a target point, e.g., in face perception, the target would be another person's face, while potential AOIs would be the eye region or mouth. Researchers have taken this approach to study, for example, the gaze behavior of participants with (sub-)clinical social anxiety or autism [11, 15, 16]. For these applications in potentially interactive scenarios, automatic procedures to generate AOIs seem beneficial [13], as manual generation can be time-consuming depending on the recording duration. In addition, the uniformity of target stimuli in these setups allow the use of standardized, published procedures (e.g. Limited-Radius Voronoi-Tessellation method, LRVT; [17]), thus ensuring the comparability of research results. In this method, AOI size can be easily regulated by adjusting the limiting radius.

The impact of AOI size on gaze classification has been investigated [13, 14]. In those studies, suggestions were made that AOIs on sparse stimuli should be large enough to be robust to noise [13], and that oversized AOIs are problematic when considering falsely positive classified gaze data [14]. While the problem of inadequate AOI sizes seems to be known, guidelines on how to choose the most appropriate size of AOIs remain vague. In addition, it is conceivable that the choice of AOI sizes depends on the gaze data's accuracy. In general, we can assume that low accuracy would require larger AOIs to ensure valid classified gaze points. However, it is unclear how accuracy and AOI size interact in affecting classification performance, and whether false-negatives and false-positives are affected differentially.

With the current study, we aimed to investigate the gaze classification performance depending on the accuracy of the detected gaze position (spatial offset between detected and real gaze position) and the AOI size (LRVT with different radii) with simulated gaze data in order to derive guidelines for selecting AOIs and their size in interactive (face-to-face) eye tracking applications. Thereby, we focus on classification performance with respect to false-positives (falsely classified inside a specific AOI; Type I error; false alarms) and false-negatives (falsely classified outside a specific AOI; Type II error; misses) to derive recommendations for choosing AOI size depending on accuracy. Specifically, we expected accuracy and AOI size to independently influence classification performance, and for the two factors to interact such that AOI size would demonstrate a greater impact on classification performance when accuracy is low. Along with these recommendations, we present a software tool that enables gaze data to be generated with a given accuracy, the visualization of gaze data on AOIs of different sizes, and the evaluation of the resulting classification of gaze data.

## Methods

### Stimuli

Facial stimuli from the Face Research Lab London (version 3; [18]) served as the basis for this simulation. We selected the following four stimuli to represent different ethnic groups: 005 (male, Asian, 28 years old), 012 (male, white, 24 years old), 025 (female, African American, 21 years old) and 134 (female, white, 21 years old). A picture with a neutral facial expression and direct gaze was chosen for each stimulus. Stimuli were resized and rescaled to 480 x 480px resembling the size of a real person sitting at a viewing distance of approx. 130cm recorded at 1920 x 1080px resolution. After rescaling, the facial stimuli covered an average area of 7.8 by 5.7˚, which corresponds to the size of a real face in a face-to-face conversation at the aforementioned viewing distance of approx. 130cm. For display in the figures, we created another stimulus that was not used for the simulation. The individual pictured in Fig 1 and Figs 3 to 5 has provided written informed consent (as outlined in PLOS consent form) to publish their image alongside the manuscript.

## a) Gaze data simulation

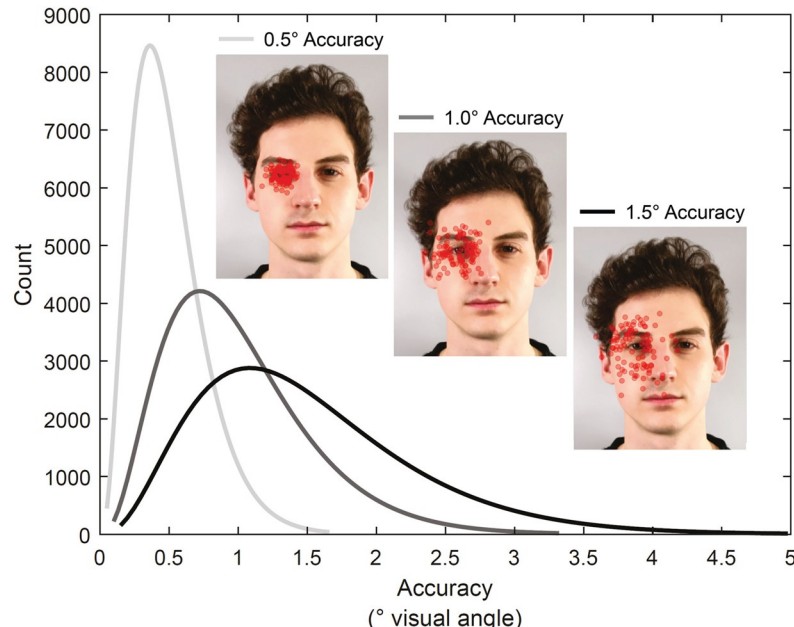

## b) Automatic AOI construction process

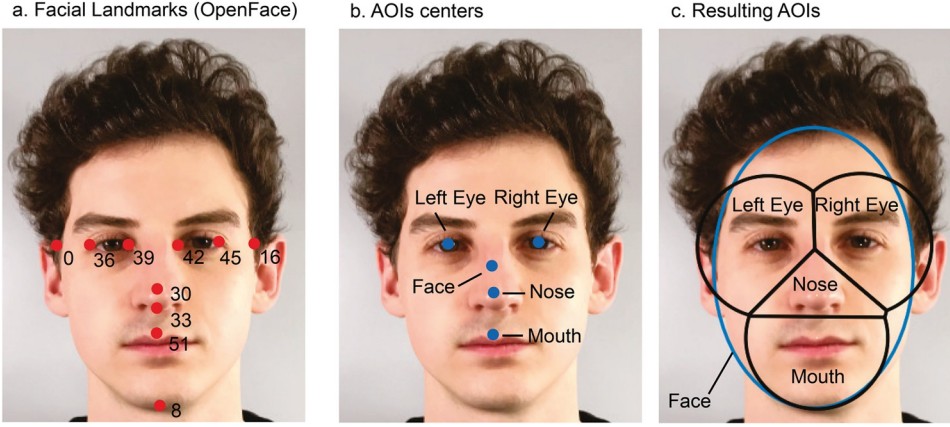

**Fig 1. Visualization of gaze data simulation and areas of interest (AOI) definition.** (A) Visualization of the three gamma functions used to generate gaze data with three levels of accuracy (0.5˚, 1.0˚ & 1.5˚) and examples of the simulated fixations for the left eye of a facial stimulus as the facial target. Each red dot represents the averaged fixation location of 30 simulated fixations; a total of n = 100 data sets were simulated. (B) Visualization of the three steps of the automatic AOI construction process. 1. Facial landmarks from OpenFace. 2. AOI center points derived from the facial landmarks. 3. Resulting AOIs using the Limited-Radius Voronoi-Tessellation (LRVT) method (example with 2.0˚ radius). Note OF = OpenFace [19]. The stimulus shown in A and B was created for illustrative purposes only and is not part of the stimulus set used in the study (see Methods section).

### Gaze data simulation

The goal for our gaze data simulation was to mimic a standard test procedure with multiple participants and several runs of a gaze validation on facial features, i.e. the instructed sequential fixation of specific targets points on a facial stimulus. For each target, fixations lasting one second each were simulated at a 120 Hz recording frequency. The coordinates of five facial features (left eye, right eye, nose, mouth & forehead) were chosen as targets. Each target point corresponded to an AOI center point, except for the forehead point, for which no AOI was

generated. Both target points and AOI center points were determined using OpenFace landmarks (Fig 1B; [19]).

To simulate a realistic gaze data set for a group of (simulated) participants, the fixation points around the facial targets were determined in four steps. (1) Mean accuracy, sample size and number of runs were specified. (2) Each simulated participant was assigned a base accuracy derived from a generalized gamma distribution around the specified mean accuracy. The standard deviation was set to 0.5 times the mean accuracy and the skewness to 0.6. (3) A random offset angle around the target point was chosen for each simulated participant. (4) Offsets per target were created for each simulated participant depending on the number of runs by varying the individual base accuracy according to a normal distribution with a standard deviation of 0.15 times the base accuracy. Runs with accuracy values that fell outside the three standard deviations were recalculated. This procedure allowed us to account for within- (Step 4) and between-subjects (Steps 1–3) variance. We applied the above-mentioned method to simulate data with mean accuracy values of 0.5˚, 1.0˚ and 1.5˚. The distribution of the simulated gaze data for the three accuracy levels is found in Fig 1A.

A total of 100 participants were simulated with 30 face validation runs for the three accuracy values and four stimuli, resulting in 36000 data sets. Each face validation run consists of a fixation for each facial target point computed by averaging the gaze samples from one second of recording at a frequency of 120 Hz. Data was simulated with an in-house tool written in Python 3.7. The tool can be downloaded here: https://osf.io/ytrh8/

## Definition of AOIs

We used the LRVT method [13] with different radii for facial features and a face ellipse to automatically define AOIs and vary their size. In the first step, facial landmarks (eyes, nose & mouth) were obtained using OpenFace [19]–Fig 1B. Second, AOI centers were either derived directly from the facial landmarks (nose & mouth) or by calculating the midpoint between two landmarks (left & right eye). The face ellipse's center was created by calculating the midpoint between the left and right eye corners (x-coordinate) and 1.5 times the distance between facial landmark 8 and 33 (y-coordinate)–Fig 1B. In the final step, AOIs were defined by applying the LRVT method with three different radii (1.0, 1.5 & 2.0˚) for the facial features and OpenFace landmarks were used to define the face ellipse. The ellipse's horizontal radius is the smaller distance between the face's center point and facial landmark 0 or 16 (x-coordinate). The vertical radius is the distance between the face center and facial landmark 8 (y-coordinate). To assess the effect of accuracy and AOI size on gaze classification performance, the AOI radius of 4˚ proposed in the literature as being robust to noise (imprecision of the signal) [13, 17] was adjusted to a 130cm viewing distance, resulting in a radius of approx. 2.0˚ (~4.6cm). This was necessary to ensure that the AOIs covered the same facial area. Additionally, this radius was reduced again twice (1.0˚ [~3.4cm] & 1.5˚ [~2.3cm]) to test the effect of different AOI sizes on classification performance.

## Data analyses

As a prerequisite for aggregating the fixations across stimuli, we performed a two-way analysis of variance (ANOVA) with the between-subject factors stimulus (005, 012, 025 & 134) and accuracy (0.5˚, 1.0˚ & 1.5˚), to test whether the percentage of correctly classified fixation points differed as a function of facial stimulus used for the simulation.

To investigate the influence of accuracy and AOI size (LRVT with different radii) on false-negatives (Type II error; misses), we analyzed the number of fixation points directed to one of the four facial AOIs that were misclassified as belonging to a different AOI, or to no AOI at all

(*rest of face* & *surrounding*). We chose to visualize the effect using confusion matrices and bar plots, and analyzed the gaze data descriptively.

The effect of accuracy and AOI size on false-positives (Type I error; false alarms) was tested by simulating fixation points on the forehead of the facial stimuli for which no AOI had been defined. Classification was correct when no AOI was detected, whereas false-positives occurred when fixations points were misclassified as belonging to one of the AOIs. Again, bar plots were created to visualize the effect of the independent variables, and analyses performed at the descriptive level.

Last, the effect of accuracy on false-negatives (Type-II error; misses) was further tested by analyzing fixation points simulated on the different AOIs as being directed towards or away from the face. Classification was correct when the fixations were detected within the face ellipse.

## Results

The two-way ANOVA for the percentage of correctly classified fixation points revealed a non-significant main effect of facial stimulus, $F(3, 1188) = 2.06$, $p = .104$, $^2_G < .01$, a significant main effect of accuracy, $F(2, 1188) = 275.77$, $p < .001$, $^2_G = .32$ and a non-significant interaction effect, $F(6, 1188) = 0.37$, $p = .900$, $^2_G < .01$, resulting in the aggregation of classification data across stimuli.

### Classification of fixations on facial feature AOIs

Fixations simulated on eyes, nose, and mouth with high quality, e.g., accuracy values of 0.5˚, were correctly classified in 96.7 to 100.0 percent of cases. Misclassification and non-classification of fixations (false-negatives; Type II error; misses), in turn, occurred in only 0 to 3.4 percent of cases. In this condition, the classification performance was largely independent of AOI size (Fig 2 right column along the vertical axis). With reduced accuracy, we noted that differences emerged in the classification performance as a function of AOI size (Fig 2 left and middle column along the vertical axis). Fixations with an accuracy of 1.5˚ were on average correctly classified in 33.4 to 79.8 percent of the cases. Large AOIs with a radius of 2.0˚ resulted in correct classification above the chance level for most AOIs (left eye, right eye & mouth), whereas classification of fixation points on the nose AOI were more evenly distributed across all other AOIs, resulting in the highest percentages of false-negatives (12.9 to 48.4%). Reducing AOI size, on the other hand, is associated with fewer misclassified fixation points, at the expense of an increase in false-negatives in terms of non-classification (rest of face & surrounding: 0 to 43.0%).

In the second step, we averaged the classification performance over all facial target points (Fig 3B). Concerning false-negative classifications, AOI size seems irrelevant for accuracy values below or equal to 0.5˚. For scenarios with accuracy values over 1.0˚, larger AOIs result in more correctly classified fixation points at the expense of a slightly increased percentage of misclassified fixation points. Smaller AOIs, on the other hand, lead to fewer misclassified fixation points, but also to a slight reduction in correctly classified fixation points attributable to 35% unclassified data-quality-dependent fixation points.

### Classification of fixations outside AOIs

To investigate the effect of accuracy and AOI size on false-positives (Type I error; false alarms), gaze data were simulated on a target point for which no AOI had been defined. In this particular case, we simulated fixations on the forehead of the facial stimuli to recreate a situation in which someone tries to avoid eye contact by hiding their behavior and fixating on the

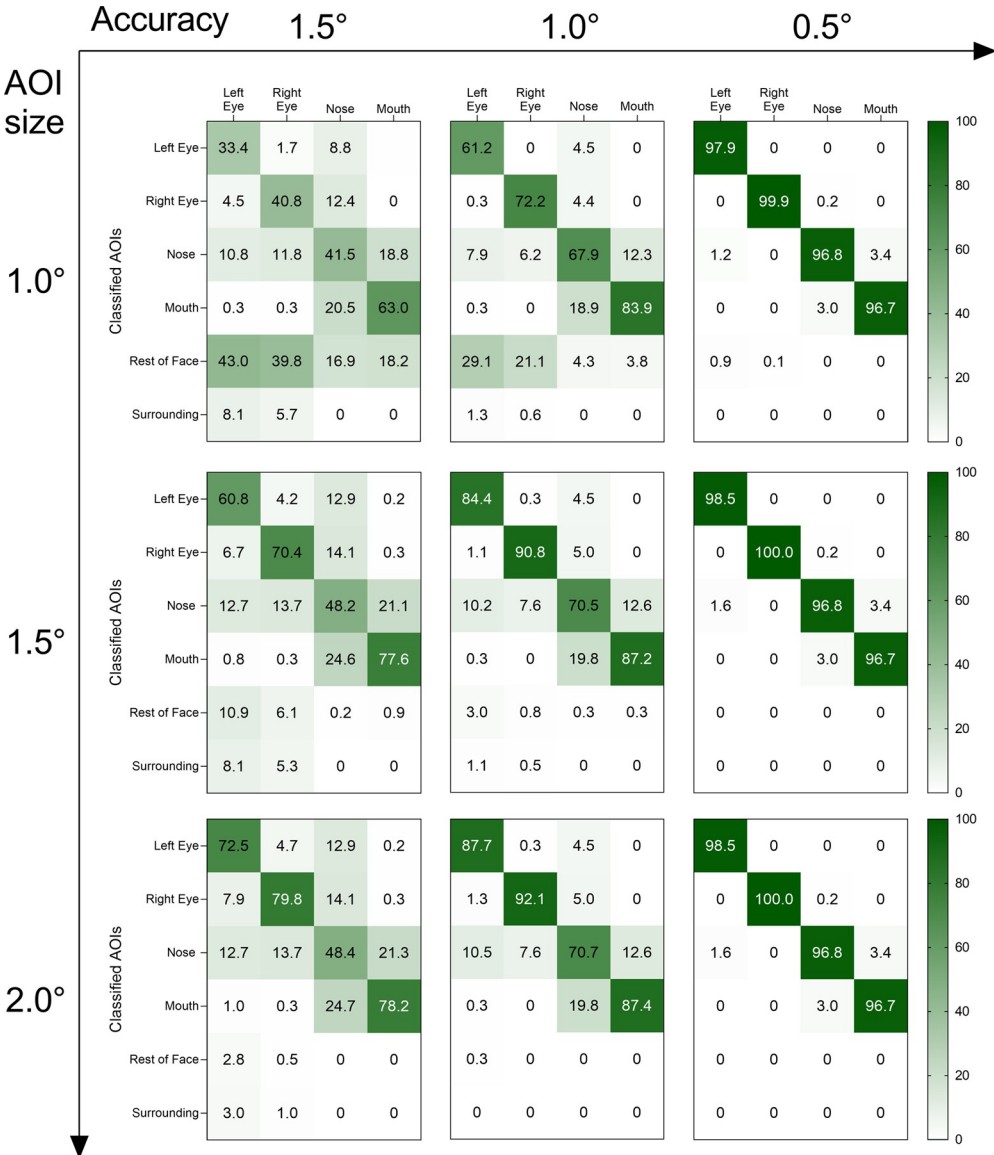

**Fig 2. Confusion matrices of the fixation classification performance as a function of gaze data accuracy and AOI size.** The percentages in the diagonal represent correctly classified fixation points, while the percentages outside the diagonal represent misclassified fixation points. The percentages in the last two rows correspond to the number of unclassified fixation points (rest of face & surrounding).

forehead. The mean distance between the forehead target point and AOI center points of the eyes was approx. 1.56˚. Fig 3D shows that large AOIs in combination with good gaze data quality (0.5˚ accuracy) result in almost all fixation points being misclassified (inflation of Type I error). As expected, most fixation points were falsely classified as belonging to the left or right eye. Smaller AOIs in combination with good data quality, on the other hand, lower numbers of false-positives drastically with over 90% of fixation points being correctly classified outside any specific AOI. The effect of accuracy is inverse, while smaller AOIs profit from better accuracy in terms of Type I errors, whereas the error increases slightly with better accuracy for larger AOIs.

Classification for gaze inside AOIs

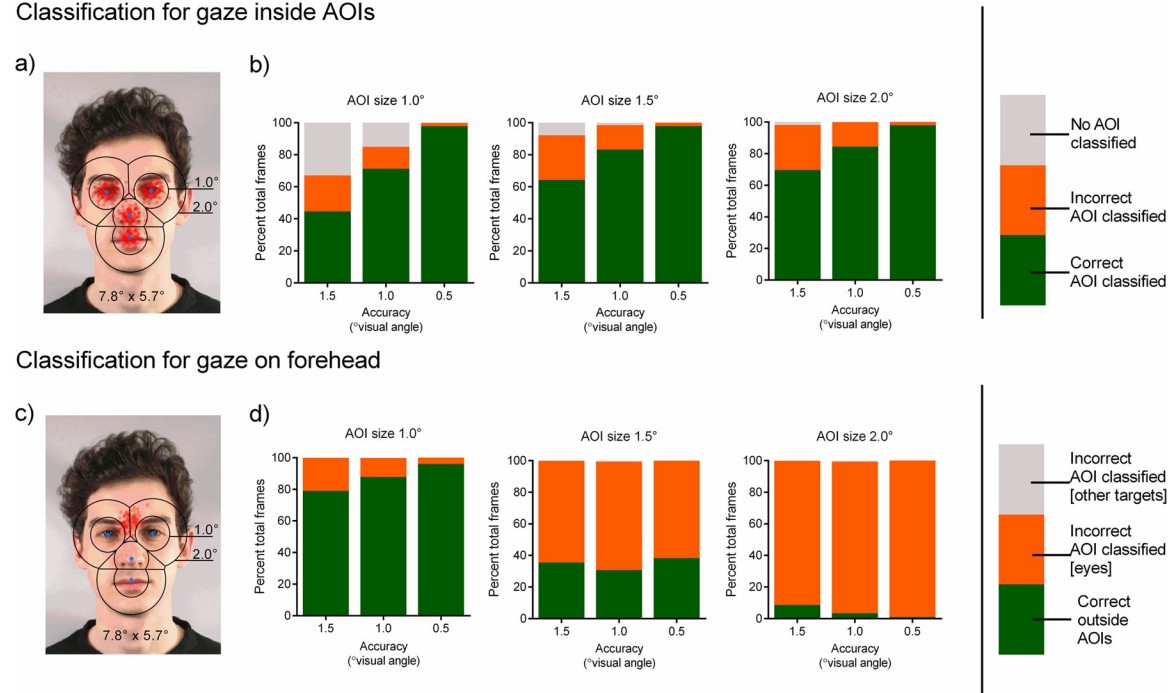

**Fig 3. Effect of gaze data accuracy and AOI size on classification performance of simulated fixations (on eyes, nose & mouth) averaged over all facial AOIs.** (A) Visualization of the AOI sizes 1.0˚ and 2.0˚ drawn around the blue center points and the simulated fixation points in red (n = 100, averaged over 30 simulated fixations; accuracy 0.5˚). (B) Green indicates correct classification within the corresponding AOI, orange and gray indicate misclassifications as fixations within the other AOIs, or no AOI at all. Effect of gaze data accuracy and AOI size on classification performance of simulated fixations (on the forehead). (C) Visualization of AOIs with 1.0˚ and 2.0˚ radius and the simulated fixation points on the forehead in red (n = 100, averaged over 30 simulated fixations; accuracy 0.5˚). (D) Green indicates the correct classification outside any AOI, orange and gray indicate misclassifications as fixations within the AOIs of the eye region or within the other AOIs. The stimulus shown in A and C was created for illustrative purposes only and is not part of the stimulus set used in the study (see Methods section).

## Classification of fixations on the face

The largest meaningful AOI for facial stimuli can be assumed to be the whole face, represented by an ellipse drawn around the entire face with no specific facial features discriminated. To investigate the effect of accuracy on false-negatives (misses), gaze data was simulated on four facial targets points (left eye, right eye, nose & mouth) and classified as being directed towards or away from the face. The face ellipse had an average vertical radius of about 4.22˚ and a horizontal radius of 2.80˚. Accuracy values up to a 1˚ degree visual angle allow nearly error-free classification of fixation points (0.5˚ accuracy: 100% correct classification; 1.0˚ accuracy: approx. 99% correct classification), but even accuracy values of a 1.5˚ degree visual angle reduce the classification performance by only about 5% (approx. 95% correct classification) (Fig 4B).

## Discussion

In the present simulation study, we investigated the effect of the accuracy of the detected gaze position and the AOI size (LRVT with different radii) on gaze data classification for facial stimuli in a (simulated) interactive eye tracking setup. As hypothesized, we found that the AOI size's effect on classification performance depend strongly on accuracy. Differentiating this effect in terms of Type I (false alarms) and Type II errors (misses), we found that AOI size is irrelevant for Type II errors when the accuracy is better than 1.0˚. The picture changed when

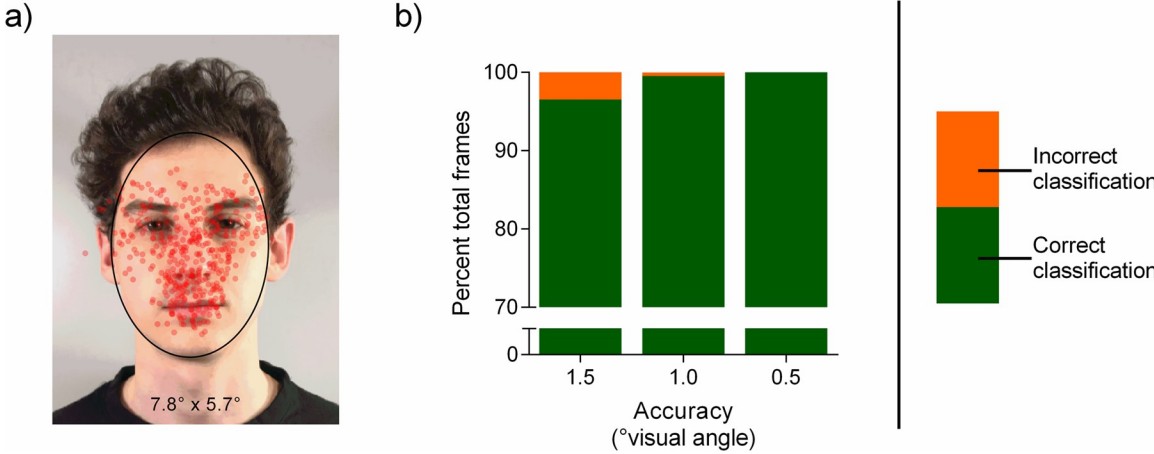

**Fig 4. Effect of gaze data accuracy on classification performance of simulated fixations (on eyes, nose & mouth) within the face ellipse.** (A) Visualization of the face ellipse and the simulated fixation points with an accuracy of 1.5° (n = 100, averaged over 30 simulated fixations) in red. (B) Green indicates correct classification within the face ellipse compared to orange for misclassification outside the face ellipse. The stimulus shown in A was created for illustrative purposes only and is not part of the stimulus set used in the study (see Methods section).

accuracy exceeded the threshold of 1.0° within the present setup: larger AOIs raised classification accuracy leading to fewer Type II errors. On the other hand, if we consider the Type I error (falsely classified inside a predefined AOI), the definition of smaller AOIs seems appropriate for all the levels of accuracy simulated here. However, the advantages of smaller AOIs decline slightly with an accuracy worse than 1.0°. Here the probability of misclassification or Type I errors increases.

Our findings regarding Type II error are consistent with previous accounts of the interaction between accuracy and AOI size on gaze data classification performance [20, 21]. Moreover, a systematic observation of gaze data classification performance on facial stimuli using the same AOI definition as in the present study concluded that larger AOIs are a noise-robust solution [13]. Therefore, larger AOIs might be a better choice with accuracy worse than 1.0°, but larger AOIs are also associated with more Type I errors (false alarms) when accuracy is low.

This distinction seems particularly relevant when examining clinical disorders such as social phobia or autism with such interactive eye tracking setups, for which small spatial differences in attention allocation are characteristic [22, 23]. We simulated such a scenario by including fixations directed at a target point on the forehead of the facial stimuli for which no AOI had been defined. We chose this scenario because people suffering from social interaction disorders often report employing strategies to normalize their gaze behavior in social interactions, such as looking between the eyes or at their interaction partner's forehead. Our simulation results are in line with another publication that recommended smaller AOIs to prevent inflated Type I errors [14]. However, by examining a wide range of accuracy values in the current study, we additionally observed that the superiority of smaller AOIs decreases in conjunction with reduced accuracy. For subject samples where abnormal gaze behavior is expected, the Type I error problem could be transformed into a Type II error problem by adding an AOI on the forehead.

Given the differential effects on Type I and Type II errors, medium-sized AOIs would represent a compromise, but when we consider the accuracy for values worse than 1.0°, on average only about 50% of fixations were classified correctly in both simulations. We therefore,

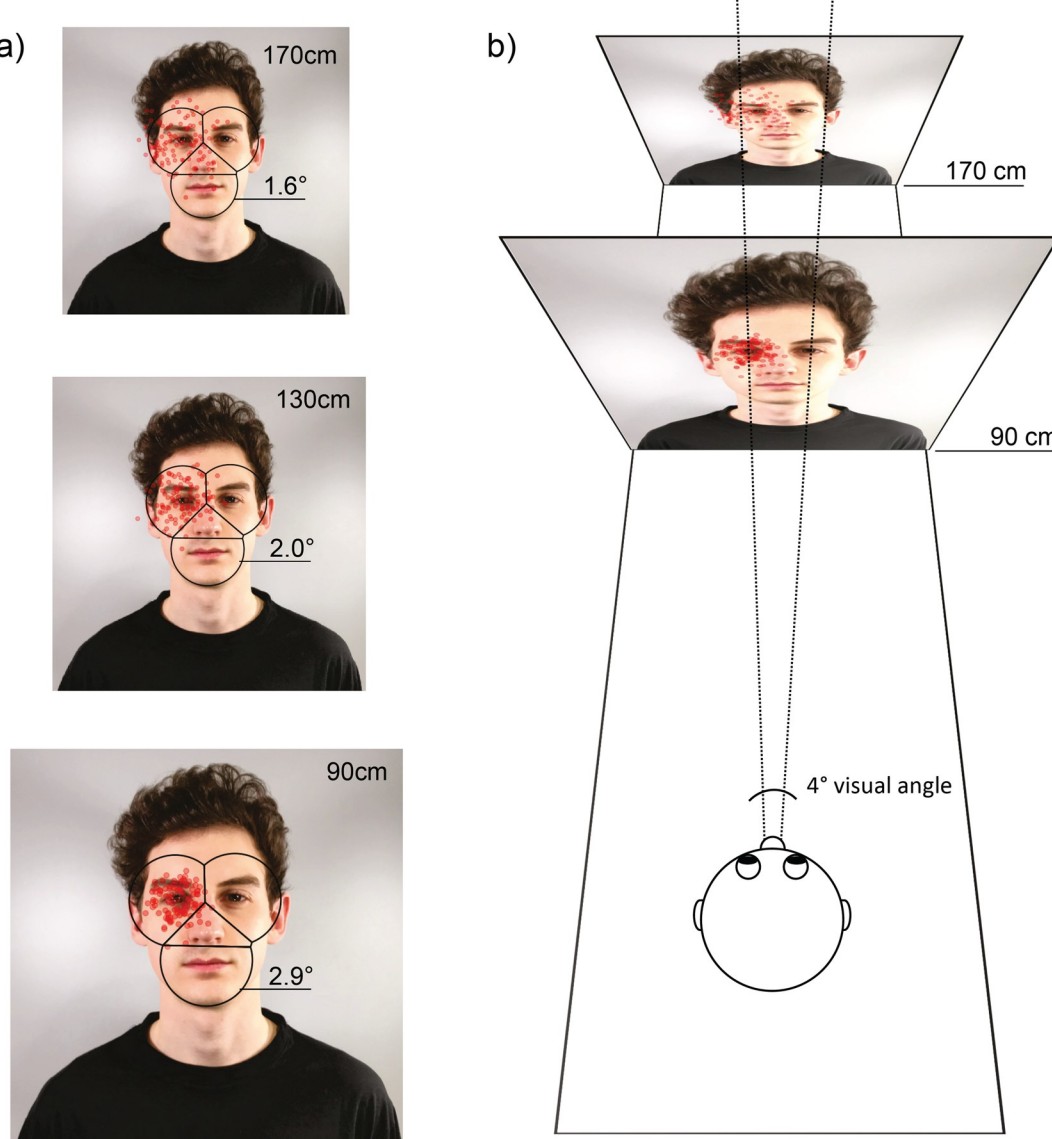

**Fig 5. Effect of viewing distance on stimulus size and fixation point deviation.** (A) Visualization of the stimulus size over three viewing distances (90cm, 130cm & 170cm) with AOIs covering the same facial areas (radius 4.6cm; in visual degree angle: 1.6˚, 2.0˚ & 2.9˚) and simulated fixation points on the left eye. (B) Visualization of the interaction between visual angle and viewing distance on fixation point deviation. In both figures (A & B) each red dot represents the averaged fixation location of 30 simulated fixations with an accuracy of 1.5˚; a total of n = 100 data set were simulated. The stimulus shown in A and B was created for illustrative purposes only and is not part of the stimulus set used in the study (see Methods section).

included a simulation designed for scenarios with low accuracy that sacrifices facial feature distinction in favor of classification performance. While 1.5˚ accuracy resulted in a classification performance around chance level when differentiating facial features, the face ellipse as the largest meaningful AOI in face perception resulted in over 95% of the fixations being classified correctly even with accuracy as low as 1.5˚.

Based on the present simulation study, the recommendations for the choice of facial AOI size in interactive eye tracking setups can be summarized as follows:

- Inflated Type I errors (falsely classified gaze points inside an AOI; false alarms) can be prevented by using small AOIs, such as radii of 1.0°, regardless of accuracy

- As previously published and recommended, the use of larger AOIs prevents the inflation of the Type II error (falsely classified gaze points outside an AOI; misses), especially for accuracy values above 1.0°

- If both error types (Type I & Type II errors) are to be compensated for in a setup with accuracy of 1.0° or better, smaller AOIs appear to be the better choice

- If both error types (Type I & Type II errors) are to be compensated for in a setup with accuracy values worse than 1.0°, we suggest not using AOIs to distinguish facial features, but instead a face ellipse as an indicator of facial gaze

While the investigator can choose the AOI size freely, accuracy is the limiting factor. Hence, it seems crucial not only to maximize accuracy as much as possible, but furthermore to estimate accuracy based on a data set recorded under the specific conditions. Ideally, distinct trials instructing the participants to fixate predefined points in the tracking area are included in the study protocol that enable accuracy to be properly calculated (validation procedures). If this is possible, one could follow the recommendations made in this study to choose the appropriate AOI size. In situations where a proper validation procedure is unfeasible, e.g. because of limited time and resources, adjusting another factor can mitigate the impact of accuracy on classification performance: the viewing distance.

In this study, we used a viewing distance of approx. 130cm to simulate an interactive eye tracking setup with a typical viewing distance for face-to-face interactions. Two factors change when reducing the viewing distance: (1) the ratio of the facial stimulus size to the surrounding in the visual field, that is, the face covers a larger area (Fig 5A), and thereby the inter-AOI distances increase, and (2) the span of fixation point deviation due to a reduced visual angle decreases, that is, the spatial accuracy increases (Fig 5B). The first aspect is also influenced by the face's actual size, e.g., in studies with toddlers the face occupies a smaller area in the viewing field even at constant viewing distance. Here, a change in inter-AOI distance (see AOI span; [13]) affects the influence of accuracy and AOI size on classification performance. However, the effect might be small in studies with an adolescent or adult sample due to the limited variation in face sizes. We thus conclude that lower accuracy can be partially compensated for by using shorter viewing distances, provided the setup allows for this. Longer viewing distances or situations with stimuli observed at varying viewing distances, on the other hand, should be treated with caution. Note that the visual angle values used in the present study need to be adjusted to differing viewing distances for meaningful comparison between studies. For example, if the viewing distance is reduced to 90cm, the AOI sizes must be increased by 0.9° from the original 2.0° radius to cover the same facial areas (Fig 5A).

One potential study limitation is that our simulation relied on the assumption that gaze data follows a unique distribution for all generated data sets. Although, the overall parameters (deviation and skewness) largely concur with the gaze data recorded in the real setup underlying the present simulation [12], it is possible that other recording conditions, populations or interaction paradigms will lead to different gaze data distributions. Therefore, we have made the gaze data simulation tool used in this study freely available and encourage repeating or modifying the simulation process.

The fact that our evaluation was based on fixations can be considered as a further study limitation. Defining fixations eliminates the influence of precision and thus improves classification performance. This has little effect in setups with eye trackers demonstrating good to very good precision, but can lead to significantly different results in less precise eye trackers. In

such cases, fixation classification is already affected by the reduced precision [24] and thus indirectly influences the gaze classification performance.

Furthermore, one might question the use of AOIs in general due to their subjective nature and the fact that the definition can be very time-consuming especially with moving AOIs in interactive setups [25]. We tried to minimize these practical disadvantages by employing a validated automatic AOI-definition method that takes a Voronoi-Tessellation approach to avoid subjectivity, and the OpenFace tool to save time. Nevertheless, data-driven approaches based on neural networks [26, 27] are a potential alternative approach when gaze data quality is uncertain or difficult to estimate, an approach that could be evaluated in future studies when used in interactive eye tracking setups.

The present study thus might help to further improve reporting standards in eye tracking research [20, 21, 28]. Furthermore, the freely available simulation tool used in the present study can be used to support the validation process of novel eye tracking setups.

Eye tracking during real social interactions is a powerful tool to examine social attention and behavior in both healthy and clinical populations. Accuracy in these setups can be compromised by movements caused by speech, facial expressions, or head rotations. It is thus essential to validate novel interactive eye tracking setups carefully. The proper estimation of accuracy is an important prerequisite for informed decisions regarding the size of AOIs used to analyze data. The results of the present simulation study indicate that smaller AOIs minimize false classifications (both Type I and Type II errors) as long as the accuracy is sufficient. For studies with lower accuracy, Type II (misses) errors can still be compensated to some extent by using larger AOIs, but at the cost of making Type I errors (false alarms) more likely. When accuracy is low, facial feature discrimination is better omitted and larger AOIs, such as a face ellipse, should be preferred to enable valid AOI classification.

## Author Contributions

**Conceptualization:** Antonia Vehlen, William Standard, Gregor Domes.

**Data curation:** Antonia Vehlen.

**Formal analysis:** Antonia Vehlen.

**Software:** Antonia Vehlen, William Standard.

**Visualization:** Antonia Vehlen.

**Writing – original draft:** Antonia Vehlen, William Standard, Gregor Domes.

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
