## [Decision Letter · Decision Letter 0]

6 Oct 2021

PONE-D-21-24588Computer-generated facial areas of interest in eye tracking research: A simulation studyPLOS ONE

Dear Dr. Domes,

Thank you for submitting your manuscript to PLOS ONE. After careful consideration, we feel that it has merit but does not fully meet PLOS ONE’s publication criteria as it currently stands. Therefore, we invite you to submit a revised version of the manuscript that addresses the points raised during the review process.

 Two expert reviewers have evaluated your work. Both reviewers are very positive and provide thoughtful and detailed comments that I am convinced will further strengthen your paper. I believe it will be possible to address all reviewer comments, and I look forward to receiving your revised manuscript. 

We look forward to receiving your revised manuscript.

Kind regards,

Guido Maiello

Academic Editor

PLOS ONE

2. Please note that according to our submission guidelines (http://journals.plos.org/plosone/s/submission-guidelines), outmoded terms and potentially stigmatizing labels should be changed to more current, acceptable terminology. To this effect, please change "Caucasian" to "white" or "of (western) European descent

“The study was in part supported by grants from the German Research Foundation (DO1312/5-1) and Trier University Research Priority Program “Psychobiology of Stress”, funded by the State Rhineland-Palatinate.”

We note that you have provided additional information within the Funding Section. Please note that funding information should not appear in other areas of your manuscript. We will only publish funding information present in the Funding Statement section of the online submission form.

 “The study was in part supported by grants from the German Research Foundation (DO1312/5-1) to GD and the Trier University Research Priority Program “Psychobiology of Stress”, funded by the State Rhineland-Palatinate. The funders had no role in study design, data collection and analysis, decision to publish, or preparation of the manuscript..”

4. We note that Figures 1A and 1B, 3 and 4 includes an image of a [patient / participant / in the study].

Reviewers' comments:

Reviewer's Responses to Questions

**Comments to the Author**

1. Is the manuscript technically sound, and do the data support the conclusions?

Reviewer #1: Yes

Reviewer #2: Yes

2. Has the statistical analysis been performed appropriately and rigorously? 

Reviewer #1: Yes

Reviewer #2: Yes

3. Have the authors made all data underlying the findings in their manuscript fully available?

Reviewer #1: Yes

Reviewer #2: Yes

4. Is the manuscript presented in an intelligible fashion and written in standard English?

Reviewer #1: Yes

Reviewer #2: Yes

5. Review Comments to the Author

Reviewer #1: In this simulation study, the authors investigate the relation between AOI size, accuracy of the gaze position signal and (in)correct classification of gaze to facial feature AOIs. The topic is relevant, and the simulations are sensible. I am therefore enthusiastic about the paper. I have only one real major comment (the first point below). However, I have a number of additional comments and suggestions that do not disqualify the relevance of the paper, but which I think can tremendously strengthen the paper and its impact.

1. As the authors currently phrase it, they investigate AOI size and accuracy of the gaze position signal. However, the third major 'factor' is the inter-AOI distance (or the size of the facial stimulus). That is, the recommendations on page 13 only hold for facial stimuli of the size used in the present study. The problem is scalable: if the inter-AOI distance doubles, and the inaccuracy doubles, the recommendation for AOI size also doubles. I strongly urge the authors to consider phrasing their problem as the combination of AOI size, inaccuracy, AND inter-AOI distance (e.g. operationalised using the AOI span measure in Hessels et al., 2016, BRM, but of course there are other ways of quantifying this). This has at least the following two advantages:

- The recommendations can be made more generic; i.e. as a suggestion for AOI size given both a known accuracy and inter-AOI distance.

- The authors can discuss reasonable inter-AOI distance to be expected in interactive setups, and the fact that this is often close to the inaccuracies that may be expected (in my experience with dual eye-tracking setups, inter-AOI distances of 2 deg and inaccuracies of 1-1.5 deg are not uncommon).

2. The analyses reported in Figure 3 and 4 (and the corresponding Results sections) contain 3 accuracy levels (0.5, 1.0, 1.5 deg) and (for figure 3) three AOI sizes (1.0, 1.5, and 2.0 deg). Why are these accuracy levels and AOI sizes used? Given that it is a simulation study, why not simulate the entire range from 0 deg (or 0.3 which seems to be roughly the lower limit for modern video-based eye trackers) to 4 deg (or something like twice/thrice the AOI span)? Such an analysis would allow readers to pick the AOI size for a given prop. (in)correct classification. (I understand that for the confusion matrices, it is not practical to do a continuous analysis)

3. In many instances, the authors write "data ..." for something that could be made explicit. A number of examples:

- "Data accuracy". Do the authors mean accuracy of the gaze position signal?

- "Data validity". I am unsure what this means. Data cannot be 'invalid' to me. Do the authors mean that the conclusion that one looks at a certain facial feature based on those data would be invalid?

- "Data simulation". Exactly what is meant with data here?

- "Data distribution". Do the authors mean the distribution of fixation positions? Or the distribution of the inaccuracy?

I suggest making all instances of "data ..." explicit whenever possible.

4. I am struggling to follow along with exactly what was simulated (e.g. lines 108-122). Perhaps a flowchart could help here, or at least some redundancy in the writing. Can it be made explicit here how many fixations where simulated for each AOI/stimulus/accuracy values/participants/etc.

5. Although I am trained as a psychologist and have been bombarded with the terms Type I and Type II, I always forget them. I understand the authors' choice for using these terms, but the reader (at least this one) could be helped by redundancy at some instances (e.g. double-coding it with other terms such as 'hit' or 'false alarm', or writing the explanation in parentheses).

6. The authors use the LRVT radius as an operationalisation for AOI size. This could also be reiterated at several locations in the text when the authors write AOI size. In my experience, it is not obvious that the AOI size is operationalised by a radius for novice readers.

Details in order of appearance:

Title. The title seems to misrepresent the topic: it's about choosing AOI size, not the fact that the AOIs are computer-generated. Why not something like:

- How to choose area-of-interest size for facial areas in interactive eye tracking?

- Facial areas of interest in eye-tracking research: How to choose AOI size

- (I'm sure there are many other great alternatives)

l.95-99. This seems incomplete: it depends also on the physical size of the screen on which it is displayed. This could be inferred from the size in degrees, but redundancy may be beneficial to the reader here.

l.105-106. Rephrase sentence. It seems grammatically off.

Figure 1. Is each participant represented by one red dot? If so, can this be made explicit?

l.134-136. What was the motivation for the 4 deg radius in previous research? And how was that used to choose the 2 deg radius here?

l.168. What is 'condition' here? Maybe I missed it, but here I already forgot. Can this be made explicit?

Section 3.2. This analysis depends drastically on the distance between the simulated fixation point and the AOI borders. However, this information is not given. Can this be given?

Section 3.3. The analysis here is uninformative without a description of the AOI size. Can this be made explicit?

l.258-261. It could be made clear that for studies investigating such participants groups or such viewing-strategies, it thus makes sense to consider an AOI for that particular location (or, alternatively, to take an AOI-free approach to check for such potential strategies).

Reviewer #2: see document - apparently I have to write 200 characters here. Lorem Ipsum sum, the rest I forgot. I liked the paper btw. - still eighty characters to go! Now it is only thirty characters... ok here we go

6. PLOS authors have the option to publish the peer review history of their article (what does this mean?). If published, this will include your full peer review and any attached files.

Reviewer #1: No

Reviewer #2: **Yes: **Benedikt Ehinger

---

## [Author Response · Author response to Decision Letter 0]

29 Dec 2021

Response to the editor 

2. Please note that according to our submission guidelines (http://journals.plos.org/plosone/s/submission-guidelines), outmoded terms and potentially stigmatizing labels should be changed to more current, acceptable terminology. To this effect, please change "Caucasian" to "white" or "of (western) European descent

Response: We thank the editor for pointing this out. We changed the term “Caucasian” to “white”.

“The following four stimuli were selected to represent different ethnic groups: 005 (male, Asian, 28 years old), 012 (male, white, 24 years old), 025 (female, African American, 21 years old) and 134 (female, white, 21 years old).”

“The study was in part supported by grants from the German Research Foundation (DO1312/5-1) to GD and the Trier University Research Priority Program “Psychobiology of Stress”, funded by the State Rhineland-Palatinate. The funders had no role in study design, data collection and analysis, decision to publish, or preparation of the manuscript.”

Response: We have removed the text referring to funding from the manuscript and want to stick to the current funding statement.

4. We note that Figures 1A and 1B, 3 and 4 includes an image of a [patient / participant / in the study]. As per the PLOS ONE policy (http://journals.plos.org/plosone/s/submission-guidelines#loc-human-subjects-research) on papers that include identifying, or potentially identifying, information, the individual(s) or parent(s)/guardian(s) must be informed of the terms of the PLOS open-access (CC-BY) license and provide specific permission for publication of these details under the terms of this license. Please download the Consent Form for Publication in a PLOS Journal (http://journals.plos.org/plosone/s/file?id=8ce6/plos-consent-form-english.pdf). The signed consent form should not be submitted with the manuscript, but should be securely filed in the individual's case notes. Please amend the methods section and ethics statement of the manuscript to explicitly state that the patient/participant has provided consent for publication: “The individual in this manuscript has given written informed consent (as outlined in PLOS consent form) to publish these case details”.

Response: For illustration purposes, we have created our own stimulus. The person depicted has signed the PLOS consent form. 

“For display in the figures, we created another stimulus that was not used for the simulation. The individual pictured in Fig 1 and Fig 3 to 5 has provided written informed consent (as outlined in PLOS consent form) to publish their image alongside the manuscript.” (p. 5)

Responses to the reviewers’ comments

Reviewer #1

In this simulation study, the authors investigate the relation between AOI size, accuracy of the gaze position signal and (in)correct classification of gaze to facial feature AOIs. The topic is relevant, and the simulations are sensible. I am therefore enthusiastic about the paper. I have only one real major comment (the first point below). However, I have a number of additional comments and suggestions that do not disqualify the relevance of the paper, but which I think can tremendously strengthen the paper and its impact.

Major comment:

1. As the authors currently phrase it, they investigate AOI size and accuracy of the gaze position signal. However, the third major 'factor' is the inter-AOI distance (or the size of the facial stimulus). That is, the recommendations on page 13 only hold for facial stimuli of the size used in the present study. The problem is scalable: if the inter-AOI distance doubles, and the inaccuracy doubles, the recommendation for AOI size also doubles. I strongly urge the authors to consider phrasing their problem as the combination of AOI size, inaccuracy, AND inter-AOI distance (e.g. operationalised using the AOI span measure in Hessels et al., 2016, BRM, but of course there are other ways of quantifying this). This has at least the following two advantages:

• The recommendations can be made more generic; i.e. as a suggestion for AOI size given both a known accuracy and inter-AOI distance.

• The authors can discuss reasonable inter-AOI distance to be expected in interactive setups, and the fact that this is often close to the inaccuracies that may be expected (in my experience with dual eye-tracking setups, inter-AOI distances of 2 deg and inaccuracies of 1-1.5 deg are not uncommon).

Response: We are very grateful for the reviewer’s comment, as it enabled us to make the influence of inter-AOI distance and the viewing distance clearer. We extended the discussion and added a figure to visualize the effect driven by two different phenomena.

“While the investigator can choose the AOI size freely, accuracy is the limiting factor. Hence, it seems crucial not only to maximize accuracy as much as possible, but furthermore to estimate accuracy based on a data set recorded under the specific conditions. Ideally, distinct trials instructing the participants to fixate predefined points in the tracking area are included in the study protocol that allow the proper calculation of data accuracy (validation procedures). If this is possible, one could follow the recommendations made in this study to choose the appropriate AOI size. In situations where a proper validation procedure is unfeasible, because of limited time and resources, adjusting another factor can mitigate the impact of accuracy on classification performance: the viewing distance.

In this study, we used a viewing distance of approx. 130cm to simulate an interactive eye tracking setup with a typical viewing distance for face-to-face interactions. Two factors change when reducing the viewing distance: (1) the ratio of the facial stimulus size to the surrounding in the visual field, that is, the face covers a larger area (Fig 5A) and thereby the inter-AOI distances increase, and (2) the span of fixation point deviation due to a reduced visual angle decreases, that is, the spatial accuracy increases (Fig 5B). The first aspect is also influenced by the face’s actual size, e.g., in studies with toddlers the face occupies a smaller area in the viewing field even at constant viewing distance. Here, a change in inter-AOI distance (see AOI span; 13) affects the influence of accuracy and AOI size on classification performance. However, the effect might be small in studies with an adolescent or adult sample due to the limited variation in face sizes. We thus conclude that lower accuracy can be partially compensated for by using shorter viewing distances, provided the setup allows for this. Longer viewing distances or situations with stimuli observed at varying viewing distances, on the other hand, should be treated with caution. Note that the visual angle values used in our study need to be adjusted to differing viewing distances for meaningful comparison between studies. For example, if the viewing distance is reduced to 90cm, the AOI sizes must be increased by 0.9° from the original 2.0° radius to cover the same facial areas (Fig 5A). 

[Fig 5]

Fig 5. Effect of viewing distance on stimulus size and fixation point deviation. (A) Visualization of the stimulus size over three viewing distances (90cm, 130cm & 170cm) with AOIs covering the same facial areas (radius 4.6cm; in visual degree angle: 1.6°, 2.0° & 2.9°) and simulated fixation points on the left eye. (B) Visualization of the interaction between visual angle and viewing distance on fixation point deviation. In both figures (A & B) each red dot represents the averaged fixation location of 30 simulated fixations with an accuracy of 1.5°; a total of n=100 data sets were simulated.” (p. 14-15)

Additional comments:

1. The analyses reported in Figure 3 and 4 (and the corresponding Results sections) contain 3 accuracy levels (0.5, 1.0, 1.5 deg) and (for figure 3) three AOI sizes (1.0, 1.5, and 2.0 deg). Why are these accuracy levels and AOI sizes used? Given that it is a simulation study, why not simulate the entire range from 0 deg (or 0.3 which seems to be roughly the lower limit for modern video-based eye trackers) to 4 deg (or something like twice/thrice the AOI span)? Such an analysis would allow readers to pick the AOI size for a given prop. (in)correct classification. (I understand that for the confusion matrices, it is not practical to do a continuous analysis).

Response: We thank the reviewer for pointing out this opportunity to extend our simulation results. We actually did this at some point when conceptualizing this paper, but refrained from reporting it. We found that the effects of accuracy levels <0.5° and >1.5° on gaze classification performance are marginal, and thus focused on an accuracy range with the greatest effect on gaze classification. If the reader experiences accuracy levels that are not reported in our study, we recommend using the gaze simulation tool published with the study. The tool can be used to adjust parameters such as the desired accuracy and to observe the effects on gaze classification performance. 

2. In many instances, the authors write "data ..." for something that could be made explicit. A number of examples:

• "Data accuracy". Do the authors mean accuracy of the gaze position signal?

• "Data validity". I am unsure what this means. Data cannot be 'invalid' to me. Do the authors mean that the conclusion that one looks at a certain facial feature based on those data would be invalid?

• "Data simulation". Exactly what is meant with data here?

• "Data distribution". Do the authors mean the distribution of fixation positions? Or the distribution of the inaccuracy?

• I suggest making all instances of "data ..." explicit whenever possible.

Response: We took the comment of the reviewer very seriously and now specify the definition of data as precisely as possible. Changes were made throughout the manuscript, here we list some examples:

“With the current study, we aimed to investigate the gaze classification performance depending on the accuracy of the detected gaze position (spatial offset between detected and real gaze position) and AOI size (LRVT with different radii) with simulated gaze data in order to derive guidelines for the selection AOIs and their size in interactive (face-to-face) eye tracking applications.” (p. 4)

“Since these setups differ substantially from the eye tracker manufacturer’s test conditions, validation is essential with regard to data quality the quality of gaze data and other factors potentially threatening the data validity validity of this signal.” (p. 2)

“The goal for our gaze data simulation was to mimic a standard test procedure with multiple participants and several runs of a gaze validation on facial features, i.e. the instructed sequential fixation of specific targets points on a facial stimulus.” (p. 5)

“Although, the overall parameters (deviation and skewness) are largely in accordance with gaze data recorded in the real setup underlying the present simulation (12), it is possible that other recording conditions, populations or interaction paradigms will lead to different gaze data distributions.” (p. 15)

3. I am struggling to follow along with exactly what was simulated (e.g. lines 108-122). Perhaps a flowchart could help here, or at least some redundancy in the writing. Can it be made explicit here how many fixations where simulated for each AOI/stimulus/accuracy values/participants/etc.

Response: We thank the reviewer for this important remark. We now describe the procedure in more detail to make it easier to understand.

“To simulate a realistic gaze data set for a group of (simulated) participants, the fixation points around the facial targets were determined in four steps. (1) Mean accuracy, sample size and number of runs were specified. (2) Each simulated participant was assigned a base accuracy derived from a generalized gamma distribution around the specified mean accuracy. The standard deviation was set to 0.5 times the mean accuracy and the skewness to 0.6. (3) A random offset angle around the target point was chosen for each simulated participant. (4) Offsets per target were created for each simulated participant depending on the number of runs by varying the individual base accuracy according to a normal distribution with a standard deviation of 0.15 times the base accuracy. Runs with accuracy values that fell outside three standard deviations were recalculated. This procedure allowed us to account for within- (Step 4) and between-subjects (Steps 1-3) variance. We applied the above-mentioned method to simulate data with mean accuracy values of 0.5°, 1.0° and 1.5°. The distribution of the simulated gaze data for the three accuracy levels is found in Fig 1A.

A total of 100 participants were simulated with 30 face validation runs for the three accuracy values and four stimuli, resulting in 36000 data sets. Each face validation run consists of a fixation for each facial target point computed by averaging the gaze samples from one second of recording at a frequency of 120 Hz.” (p. 5-6)

4. Although I am trained as a psychologist and have been bombarded with the terms Type I and Type II, I always forget them. I understand the authors' choice for using these terms, but the reader (at least this one) could be helped by redundancy at some instances (e.g. double-coding it with other terms such as 'hit' or 'false alarm', or writing the explanation in parentheses).

Response: We thank the reviewer for bringing this problem to our attention. We now include more redundancy in the text.

“In addition, these effects were not independent and differed for falsely classified gaze inside AOIs (Type I errors; false alarms) and falsely classified gaze outside the predefined AOIs (Type II errors; misses).” (p.2)

“Thereby, we focus on classification performance with respect to false-positives (falsely classified inside a specific AOI; Type I error; false alarms) and false-negatives (falsely classified outside a specific AOI; Type II error; misses) to derive recommendations for choosing AOI size depending on accuracy.” (p.4)

“To investigate the influence of accuracy and AOI size (LRVT with different radii) on false-negatives (Type II error; misses), we analyzed the number of fixation points directed to one of the four facial AOIs that were misclassified as belonging to a different AOI, or to no AOI at all (rest of face & surrounding). We chose to visualize the effect using confusion matrices and bar plots, and analyzed the gaze data descriptively. 

The effect of accuracy and AOI size on false-positives (Type I error; false alarms) was tested by simulating fixation points on the forehead of the facial stimuli for which no AOI had been defined. Classification was correct when no AOI was detected, whereas false-positives occurred when fixations points were misclassified as belonging to one of the AOIs. Again, bar plots were created to visualize the effect of the independent variables, and analyses were performed at the descriptive level. 

Last, the effect of accuracy on false-negatives (Type-II error; misses) was further tested by analyzing fixation points simulated on the different AOIs as being directed towards or away from the face. Classification was correct when the fixations were detected within the face ellipse.” (p.8)

“Fixations simulated on eyes, nose, and mouth with high data quality, e.g., accuracy values of 0.5°, were correctly classified in 96.7 to 100.0 percent of cases. Misclassification and non-classification of fixations (false-negatives; Type II error; misses), in turn, occurred in only 0 to 3.4 percent of cases. (p.9)

“To investigate the effect of accuracy and AOI size on false-positives (Type I error; false alarms), gaze data were simulated on a target point for which no AOI had been defined. In this particular case, we simulated fixations on the forehead of the facial stimuli to recreate a situation in which someone is trying to avoid eye contact and hides their behavior by fixating on the forehead.” (p.10)

“Differentiating this effect in terms of Type I (false alarms) and Type II errors (misses), we found that AOI size is irrelevant concerning Type II error (falsely classified outside predefined AOIs) for data accuracy better than 1.0°.” (p.12)

“For studies with lower accuracy, Type II errors (misses) can still be compensated to some extent by using larger AOIs, but at the cost of increasing the probability of Type I errors (false alarms).” (p. 16)

5. The authors use the LRVT radius as an operationalisation for AOI size. This could also be reiterated at several locations in the text when the authors write AOI size. In my experience, it is not obvious that the AOI size is operationalised by a radius for novice readers.

Response: Following the reviewer's advice, we have added the information at certain points in the text.

“With the current study, we aimed to investigate the gaze classification performance depending on the accuracy of the detected gaze position (spatial offset between detected and real gaze position) and AOI size (LRVT with different radii) with simulated gaze data in order to derive guidelines for the selection AOIs and their size in interactive (face-to-face) eye tracking applications.” (p. 4)

“To investigate the influence of accuracy and AOI size (LRVT with different radii) on false-negatives (Type II error; misses), we analyzed the number of fixation points directed to one of the four facial AOIs that were misclassified as belonging to a different AOI, or to no AOI at all (rest of face & surrounding).” (p.8)

“In the present simulation study, we investigated the effect of the accuracy of the detected gaze position and the AOI size (LRVT with different radii) on gaze data classification for facial stimuli in a (simulated) interactive eye tracking setup.” (p. 12)

Details:

1. Title. The title seems to misrepresent the topic: it's about choosing AOI size, not the fact that the AOIs are computer-generated. Why not something like:

• How to choose area-of-interest size for facial areas in interactive eye tracking?

• Facial areas of interest in eye-tracking research: How to choose AOI size

• (I'm sure there are many other great alternatives)

Response: We thank the reviewer for these suggestions and have adjusted the title to read:

 “How to choose the size of facial areas of interest in interactive eye tracking”

2. L. 95-99. This seems incomplete: it depends also on the physical size of the screen on which it is displayed. This could be inferred from the size in degres, but redundancy may be beneficial to the reader here.

Response: Our gaze data were simulated from the camera's perspective to the stimuli (above the participants' heads; Vehlen et al., 2021) and are thus independent of monitor size. We did this to simulate realistic gaze data in a face-to-face setup without monitors. 

3. L.105-106. Rephrase sentence. It seems grammatically off.

Response: We have reworded the sentence to make it grammatically correct:

“Each target point corresponded to an AOI center point, except for the forehead point, for which no AOI was generated.” (p.5)

4. Figure 1. Is each participant represented by one red dot? If so, can this be made explicit?

Response: We extended the caption to figure 1:

“Visualization of the three gamma functions used to generate gaze data with three levels of accuracy (0.5°, 1.0° & 1.5°) and examples of the simulated fixations for the left eye of a facial stimulus as the facial target. Each red dot represents the averaged fixation location of 30 simulated fixations; a total of n=100 data sets were simulated. (B) Visualization of the three steps of the automatic AOI construction process. 1. Facial landmarks from OpenFace. 2. AOI center points derived from the facial landmarks. 3. Resulting AOIs using the Limited-Radius Voronoi-Tessellation (LRVT) method (example with 2.0° radius). Note OF = OpenFace (19).” (p. 7)

5. L.134-136. What was the motivation for the 4 deg radius in previous research? And how was that used to choose the 2 deg radius here?

Response: We tried to make this reference more explicit by adding the following information:

“To assess the effect of accuracy and AOI size on gaze classification performance, the AOI radius of 4° proposed in the literature as being robust to noise (imprecision of the signal) (13,17) was adjusted to a 131cm viewing distance, resulting in a radius of approx. 2.0° (~4.6cm). This was necessary to ensure that the AOIs covered the same facial area.” (p. 7)

6. L.168. What is 'condition' here? Maybe I missed it, but here I already forgot. Can this be made explicit?

Response: We thank the reviewer for pointing out this inexactness and have revised the section:

“The two-way ANOVA for the percentage of correctly classified fixation points revealed a non-significant main effect of facial stimulus, F(3, 1188) = 2.06, p = .104, ƞ2G < .01, a significant main effect of accuracy, F(2, 1188) = 275.77, p < .001, ƞ2G = .32 and a non-significant interaction effect, F(6, 1188) = 0.37, p = .900, ƞ2G < .01, resulting in the aggregation of classification data across stimuli.” (p. 8)

7. Section 3.2. This analysis depends drastically on the distance between the simulated fixation point and the AOI borders. However, this information is not given. Can this be given?

Response: We completely agree with the reviewer and added the appropriate information. In the process, we noticed an error in the specified sizes of the stimuli, which we have also corrected.

“To investigate the effect of accuracy and AOI size on false-positives (Type I error; false alarms), gaze data were simulated on a target point for which no AOI had been defined. In this particular case, we simulated fixations on the forehead of the facial stimuli to recreate a situation in which someone is trying to avoid eye contact and hides their behavior by fixating on the forehead. The mean distance between the forehead target point and the AOI center points of the eyes was approx. 1.56°.” (p. 10)

“After rescaling, the facial stimuli covered an average area of 7.8 by 5.7°, which corresponds to the size of a real face in a face-to-face conversation at the aforementioned viewing distance.” (p. 5)

8. Section 3.3. The analysis here is uninformative without a description of the AOI size. Can this be made explicit?

Response: We added the requested information.

“To investigate the effect of data accuracy on false-negatives (misses), gaze data was simulated on four facial targets points (left eye, right eye, nose & mouth) and classified as being directed towards or away from the face. The face ellipse had an average vertical radius of about 4.22° and a horizontal radius of 2.80°.” (p. 11)

9. L.258-261. It could be made clear that for studies investigating such participants groups or such viewing-strategies, it thus makes sense to consider an AOI for that particular location (or, alternatively, to take an AOI-free approach to check for such potential strategies).

Response: We extended the discussion accordingly.

“However, by examining a wide range of accuracy values in the current study, we additionally observed that the superiority of smaller AOIs decreases in conjunction with reduced accuracy. For subject samples where abnormal gaze behavior is expected, the Type I error problem could be transformed into a Type II error problem by adding an AOI on the forehead.” (p. 13)

Reviewer #2

In this study, the authors simulate fixations on faces with different accuracies (noise levels of the simulated Eye-Tracker). They then analyze how well fixation location in different ROIs can be estimated using four ROIs and three ROI-sizes. They quantify both type-1 and type-2 errors. They find the expected patterns that high accuracy is good, large ROIs lead to higher type-2 errors, but lower type-1 errors, but importantly, they give a clear framework and open-source tool on how to quantify these intuitions. Consequently, this helps in planning and intuiting eye-tracking face-perception studies with eye-trackers of different accuracy. 

As the authors state in the paper, a limitation is, that the ROI-size, eye-tracking accuracy and stimulus sizes are all connected via viewing distance, and one can be traded off each other’s. A limitation in their type-2 error analysis is, that it hinges heavily on the placement of the “wrong” fixation point (in this case the forehead). But this is clearly visible from the paper. I encourage the authors to polish their tool a bit, because I think it could be very useful for the community.

I enjoyed reading this paper and I have only some minor comments.

Weaknesses:

1. Code not available

Response: 

We thank the reviewer for this suggestion and uploaded the code to Zenodo (https://doi.org/10.5281/zenodo.5750368) and github (https://github.com/avehlen/AOI_ET). The links can also be found in the OSF repository (https://osf.io/ytrh8/). 

Minor Comments:

1. Unclear definition of “fixation point”.

As I understand from the data, you check for each simulated sample, whether it is inside or outside the AOI. I understood fixation point more as the average x/y of a fixation. “Gaze sample” could be a better term? 

Response: We thank the reviewer for pointing out this inconsistency in our results. We now based all calculations on fixations and uploaded the data to OSF. As expected, gaze classification improved across all conditions as a result of this change. We discussed this observation in the following manner:

“The fact that our evaluation was based on fixations can be seen as a further study limitation. Defining fixations eliminates the influence of precision and thus improves classification performance. This has little effect in setups with eye trackers demonstrating good to very good precision, but can lead to significantly different results in less precise eye trackers. In such cases, fixation classification is already affected by the reduced precision (24) and thus indirectly influences the gaze classification performance.” (p. 15)

2. Related to the above: Often, accuracies of eye trackers are calculated on fixation level and not on sample, thus counteracting potential influences of bad precision. It did not become clear to me, whether we should count the 120 samples as 120 Fixations, or as a single fixation.

Response: By presenting the results on fixation level, the effect of precision is eliminated. 

3. Thanks to the shared data, I reproduced some part of the main analyses of the paper. This replicates Figure 3b, but in reverse order of the x-axis and not as stacked barplots. Note that the means perfectly match Figure 3b – I additionally depicted the variability over subjects, which I think show the relation to Figure 1a, the gamma distributions of “base” accuracy”. Maybe this variability is of interest to the readers as well, quite often we aggregate too much and do not show the variability; but in this case I’m actually not sure.

Response: We thank the reviewer for taking the time to visualize the data. We believe that distribution information is in general very valuable, but for reasons of clarity, we decided to use the original type of visualization. The limited variance due to the fixation-based analysis was another factor that led to our decision.

4. It would be super valuable, if you could host the source code of the simulation tool on github (or other platform + zenodo link for archiving), so that other researchers can build upon your work. Further, there is no documentation on how to use the tool (or did I miss it?), and I couldn’t get it to run properly. If documentation is too time-consuming, a short screencap would already help.

Response: We thank the reviewer for this suggestion. We uploaded the code to Github (https://github.com/avehlen/AOI_ET) and Zenodo (https://doi.org/10.5281/zenodo.5750368). On OSF we have linked two videos describing the use of the tool (https://osf.io/ytrh8/). 

5. On a similar note, it would be nice to get a short readme file in your data.zip, simply depicting what column means what. I think I figured it out, but I was confused at first.

Response: We uploaded the new fixation-based data to OSF and added a detailed readme file. 

6. Plotting Suggesting: Maybe you can add the sizes of the stimuli (2.8°/1.9°) to one (or all?) stimulus depictions? That would help in setting the 1.5° accuracy better into relation.

Response: We did this for Figures 3 and 4.

7. L319/L320: repetition of “essential”

Response:

“It is thus essential to validate novel interactive eye tracking setups carefully. The proper estimation of accuracy is an important prerequisite for informed decisions regarding the size of AOIs used for data analysis.” (p. 15)

8. L113 was => were

Response: This sentence was removed in response to reviewer 1’s third comment. 

“Second, each simulated participant was assigned a base accuracy derived from a generalized gamma distribution around the specified mean accuracy.” (p.6)

---

## [Decision Letter · Decision Letter 1]

24 Jan 2022

How to choose the size of facial areas of interest in interactive eye tracking

PONE-D-21-24588R1

Dear Dr. Domes,

We’re pleased to inform you that your manuscript has been judged scientifically suitable for publication and will be formally accepted for publication once it meets all outstanding technical requirements.

Kind regards,

Guido Maiello

Academic Editor

PLOS ONE

Additional Editor Comments (optional):

Reviewers' comments:

Reviewer's Responses to Questions

**Comments to the Author**

1. If the authors have adequately addressed your comments raised in a previous round of review and you feel that this manuscript is now acceptable for publication, you may indicate that here to bypass the “Comments to the Author” section, enter your conflict of interest statement in the “Confidential to Editor” section, and submit your "Accept" recommendation.

Reviewer #1: All comments have been addressed

Reviewer #2: All comments have been addressed

2. Is the manuscript technically sound, and do the data support the conclusions?

Reviewer #1: (No Response)

Reviewer #2: Yes

3. Has the statistical analysis been performed appropriately and rigorously? 

Reviewer #1: (No Response)

Reviewer #2: Yes

4. Have the authors made all data underlying the findings in their manuscript fully available?

Reviewer #1: (No Response)

Reviewer #2: Yes

5. Is the manuscript presented in an intelligible fashion and written in standard English?

Reviewer #1: (No Response)

Reviewer #2: Yes

6. Review Comments to the Author

Reviewer #1: (No Response)

Reviewer #2: Thank you for adressing all my concers. This is a super nice and open manuscript. Well done, keep up the good, helpful and important work!

7. PLOS authors have the option to publish the peer review history of their article (what does this mean?). If published, this will include your full peer review and any attached files.

Reviewer #1: No

Reviewer #2: **Yes: **Benedikt Ehinger

---

## [Editor Report · Acceptance letter]

26 Jan 2022

PONE-D-21-24588R1 

How to choose the size of facial areas of interest in interactive eye tracking 

Dear Dr. Domes:

I'm pleased to inform you that your manuscript has been deemed suitable for publication in PLOS ONE. Congratulations! Your manuscript is now with our production department. 

Kind regards, 

on behalf of

Dr. Guido Maiello 

Academic Editor

PLOS ONE